# Screening Drugs for Broad-Spectrum, Host-Directed Antiviral Activity: Lessons from the Development of Probenecid for COVID-19

**DOI:** 10.3390/v15112254

**Published:** 2023-11-14

**Authors:** Ralph A. Tripp, David E. Martin

**Affiliations:** 1Department of Infectious Diseases, University of Georgia, Athens, GA 30602, USA; 2TrippBio, Inc., Jacksonville, FL 32256, USA; davidmartin@trippbio.com

**Keywords:** direct-acting antiviral (DDA), host-directed antiviral (HDA), drug discovery assays, probenecid, virus

## Abstract

In the early stages of drug discovery, researchers develop assays that are compatible with high throughput screening (HTS) and structure activity relationship (SAR) measurements. These assays are designed to evaluate the effectiveness of new and known molecular entities, typically targeting specific features within the virus. Drugs that inhibit virus replication by inhibiting a host gene or pathway are often missed because the goal is to identify active antiviral agents against known viral targets. Screening efforts should be sufficiently robust to identify all potential targets regardless of the antiviral mechanism to avoid misleading conclusions.

## 1. Introduction

Compounds that exert their activity through viral targets are known as direct-acting antivirals (DDAs). While this strategy is effective at identifying compounds with antiviral activity, the overall success rate of clinical drug development remains low at around 10% [1,2]. This raises several questions as to why 90% of drug development fails despite the implementation of many effective DDA screening strategies. Multiple factors may contribute including toxicity, poor drug-like properties, and a lack of clinical efficacy [3]. In addition to DDAs, there is another type of antiviral that has antiviral activity via the host cell rather than the virus. These are known as host-directed antivirals (HDAs). DAAs are compounds that interact with viral proteins to exert antiviral function [4], while HDAs interfere with cellular pathways [5]. Targeting host factors rather than viral components to inhibit viral replication offers the advantage of an increased threshold to viral resistance and can provide broad-spectrum antiviral action against different viruses. The discovery of antiviral compounds for SARS-CoV-2 has mainly focused on DAAs, including the three FDA-approved drugs for clinical use; namely, Remdesivir, PF-07321332 (paxlovid), and MK-4482/EIDD-2801 (molnupiravir) [6,7,8]. A further example of a DAA that targets SARS-CoV-2 is the RNA-dependent RNA polymerase (RdRp) inhibitors which target the virus and which have no human homolog. There are two types of RdRp inhibitors: nucleoside analogs and non-nucleoside analogs [9]. An example of an HDA that targets SARS-CoV-2 is piperlongumine, an alkaloid/imide extracted from the long pepper (*Piper longum*) [10] that selectively induces reactive oxygen species in infected cells by GSTP1 (Glutathione S-Transferase Theta 1) inhibition, protecting mice against SARS-CoV-2 and variants of concern. While many DAAs have been identified using in vitro drug screens, very few have been validated in preclinical studies and fewer in clinical studies. Unfortunately, fewer HDAs have been tested in preclinical or clinical studies. Various drug screening approaches have been used to identify potential DAAs and HDAs to treat virus infection, but efforts have centered on broad-based screening efforts commensurate with determining DAAs for drug discovery. Of the FDA-approved antivirals, hepatitis C virus (HCV) and human immunodeficiency virus (HIV) account for approximately two-thirds of all approvals [11], and small molecules dominate the antiviral drugs, constituting nearly 90% of the antivirals. Approximately 10% of all approved antivirals are directed against host proteins [12]. This is owing to several barriers, with one being the in-depth knowledge needed for understanding virus–host interactions and their significance to virus replication. Another barrier is the lack of a suitable model as HDAs are designed to target human genes/factors and the animal models may not truly represent the host. A range of tools to elucidate the virus–host interactions by perturbing the host cell function have aided in determining specific host genes and cellular proteins involved in viral replication. For example, genome-wide loss of function analysis using RNAi technology has unveiled key host gene functions [13,14,15,16]. The discovery of cellular factors that are critical for virus replication, but are dispensable for the host, can serve as a target for antiviral drug development. A brief list of some of the HDAs that have been investigated against viruses are shown in Table 1.

## 2. High Throughput Drug Screening (HTS)

The COVID-19 pandemic has sparked global research for effective antiviral drugs. The National Center for Advancing Translational Sciences (NCATS) has developed simple, robust early-stage drug discovery assays for DAAs, which are less complex than those used for HDA discovery. DAAs are compounds that bind to a specific target with high selectivity, affinity, and potency. These methods also can be used for mass screening against those targets to find compounds that meet these criteria. The design of high throughput screening (HTS) for HDAs is challenging due to the need for prior knowledge of virus–host interactions. This requirement makes it different from NCAT-based screens used typically by large pharmaceutical companies and organizations. HDA drugs inhibit virus replication by inhibiting a necessary host cell pathway. This is relevant in pandemic preparedness where the goal is to identify active antiviral agents; as such, screening efforts should be sufficiently robust to identify all potential targets regardless of the antiviral mechanism. Failure to identify all potential targets can lead to misleading conclusions. Simply put, just because screening methods were negative does not necessarily mean the drug was not active.

## 3. HDA Bias in the HTS Approach

An example of HDA bias in the HTS approach is demonstrated with probenecid. NCATS performed a large drug screen to evaluate compounds with antiviral activity against SARS-CoV-2 (Open Data Portal (nih.gov), Table 2). The results of the screening effort suggested that there was no anti-SARS-CoV-2 drug activity. Probenecid is a repurposed drug that is used to treat gout by preventing the kidneys from reabsorbing uric acid, which can lead to the formation of crystals in the joints [23]. In addition to its uricosuric properties, probenecid has been shown to have antiviral and anti-inflammatory properties [24]. As an antiviral drug, probenecid has been shown to potentially inhibit the SARS-CoV-2, influenza virus, as well as inhibit RSV replication in vitro and in vivo [25], and, more importantly, it has exhibited antiviral activities and some anti-inflammatory activities in a Phase 2 clinical study [24]. Preclinically, nanomolar to micromolar concentrations of probenecid have been shown to inhibit the replication of ancestral SARS-CoV-2 and variants (Beta, Gamma, Delta, and Omicron, B.1.1) in Vero E6 cells and in NHBE cells, as well as inhibit RSV and influenza replication in several respiratory epithelial cell lines [24,25,26,27]. In a dose-finding Phase 2 study of non-hospitalized patients with symptomatic, mild-to-moderate COVID-19, probenecid-treated patients showed a median time to viral clearance that was significantly shorter for 1000 mg vs. 500 mg probenecid treated patients (7 vs. 9 days, respectively; *p* < 0.0001) compared to placebo-treated patients (day 11; *p* < 0.0001) [24]. Patients treated with probenecid showed a significant, dose-dependent decrease in the time to viral clearance, and a significantly higher proportion of patients had complete symptom resolution by day 10 and a more rapid resolution of fever than placebo-treated patients. This broad antiviral activity in preclinical and clinical studies across multiple SARS-CoV-2 variants that was observed for RSV and influenza, as well as for different cell line and animal models, suggests that probenecid was affecting a common host cell pathway used by more than one virus for replication.

## 4. Drug Screening Endpoints

While drug screening assays include the measurement of host cell destruction (CPE) or host cell viability to identify antiviral agents, these assays are not suitable for drugs like probenecid that do not display cytotoxicity. The most eligible method for testing antiviral susceptibility of probenecid and other HDAs is the direct measurement of viral replication using techniques such as a virus plaque reduction assay. The antiviral activity can be measured by quantifying the percentage of infected cells either visually by counting plaques in a cell lawn or by using an imaging instrument (such as the Cell Insight CX5 (Thermo Fisher Scientific, Waltham, MA, USA) that can be used for high-content screening. The CX5 screening platform offers multiplexable wavelengths for fluorescence and white light illumination for colorimetric samples. One modest bioassay method is to determine virus plaques linked to virus infection, e.g., for SARS-CoV-2 this method uses Vero E6 cells and evaluates the inhibition of SARS-CoV-2-induced CPE or plaques as measured using crystal violet staining. 

There are various methods to evaluate antiviral HDAs that target the distinct stages in the viral life cycle, such as entry, uncoating, genome replication, genome packaging and assembly, release, and maturation. Viruses use single or multiple cell receptors for infection and enter host cells either through endocytosis or by binding to a cell surface receptor [32]. Screening for HDA drugs that inhibit viral and plasma membrane fusion is a common practice. However, it is important to note that inhibiting virus–cell fusion does not necessarily indicate the inhibition of virus replication. Therefore, the results of such screenings can be misleading. Furthermore, it is important to keep in mind that for most viruses, the activation of the fusion or penetration mechanisms occurs through conformational changes and structural rearrangements in viral surface proteins. Determining the point at which the virus capsid and/or lipid envelope is disassembled in order to free the viral genome does not necessarily indicate virus replication. It is noteworthy that RNA viruses usually replicate in the cytoplasm, while DNA viruses and some negative-stranded RNA viruses require entry into the nucleus for replication to occur [32]. During the replication of viral genomes, RNA-dependent RNA synthesis or RNA-dependent DNA synthesis (reverse transcription) takes place [32]. However, PCR-based assays that detect virus replication levels are not always reliable as they only detect genetic material, including remnants of dead viruses, and they do not necessarily indicate active infection. Moreover, false priming, poor DNA synthesis, insufficient amplification, too few PCR cycles, and PCR inhibition can negatively affect the results [33].

It is worth noting that viruses can assemble in different locations within the host cell. Some viruses assemble in contact with the cellular membrane, while others assemble in either the nucleus or cytoplasm. For instance, paramyxoviruses assemble underneath the cytoplasmic membrane, which helps in virus assembly by providing a scaffolding function [32]. On the other hand, coronaviruses assemble at the ER–Golgi intermediate compartment (ERGIC) [33]. Viruses require assembly from precursor peptides/proteins. HDAs may target virus assembly. This property of virus replication appears to be affected by probenecid expressing its activity on MAPKs and OAT3. For example, SARS-CoV-2 assembles at the ERGIC and requires efficient virion assembly by targeting the budding site and interacting with each other or the ribonucleoprotein [39]. It is known that probenecid inhibits SARS-CoV-2 and other virus replication [25], linking virus assembly as the site of inhibition of virus replication. 

## 5. Conclusions

In conclusion, in vitro screening approaches have identified drugs that are effective in vitro but fail in vivo. One example is chloroquine/hydroxychloroquine, which inhibits SARS-CoV-2 replication in vitro but has no utility in COVID-19 patients [40,41,42]. One factor that may contribute to differences is the multifaceted response of host cells to virus infection, which includes factors that can inhibit virus replication, such as interferons (IFNs). Viruses have evolved mechanisms to escape the host IFN response, thus, the timing, readout, and type of HDA assay used to assess efficacy is critical. The selectivity index of a drug that inhibits virus replication is often determined by comparing its concentration to the concentration that causes cytotoxicity in cells. However, screening of HDAs using the CPE-based assay may be limited, particularly for those that require inducing type I IFNs. In vitro screens that use Vero E6 cells would prove ineffective for such HDAs since Vero E6 cells do not produce type I IFNs [43]. Despite the various HDAs that may be used to reveal drug efficacy, no data on drug kinetics, metabolism, or toxicity essential for clinical trials are obtained from in vitro tests. Probenecid is a cautionary note as NCATS-advocated strategies for HTS were unable to identify the novel mechanism of action of this HDA as the Phase II clinical data clearly confirm the antiviral potency [24,44].

## Figures and Tables

**Table 1 viruses-15-02254-t001:** Example of HDAs against viruses.

Drug	Host Factor	Mechanism	Virus	Reference
Fludase	Sialic acid	Cleaves SA on the host cell surface.	Influenza	[17]
Verdinexor (KPT-185)	Nuclear export	Prevent nuclear export.	Influenza, HIV-1	[18]
Trametinib	MEK	Suppresses viral replication.	Influenza	[19]
Aprotinin	Endosomes	Inhibits viral protease activity, viral entry.	Influenza, SARS-CoV-2	[20]
Gefitinib (Iressa)	EGFR	Inhibits replication of HBV via downregulation of STAT3.	HBV	[21]
RGFP966	HDAC3	Inhibits Apo-A1 and LEAP-1.	HCV	[22]

Table 1 abbreviations: SA (sialic acid), EGFR (epidermal growth factor receptor), MEK (mitogen-activated protein kinase kinase), Apo-A1 (apolipoprotein A1), LEAP-1 (liver-expressed antimicrobial peptide 1), HIV-1 (human immunodeficiency virus-1) HBV (hepatitis B virus), and HCV (hepatitis C virus).

**Table 2 viruses-15-02254-t002:** Overview of the NCATS SARS-CoV-2 screening algorithm *.

Live Virus Infectivity	Reference
SARS-CoV-2 cytopathic effect (CPE)	[28]
SARS-CoV-2 cytopathic effect (host tox counter)	[28]
** In Vitro Infectivity **	
SARS-CoV-2 pseudotyped particle entry	[29]
SARS-CoV-2 pseudotyped particle entry (tox counter screen)	[29]
SARS-CoV-2 pseudotyped particle entry	[30]
SARS-CoV-2 pseudotyped particle entry (VeroE6 tox counter screen)	[30]
MERS pseudotyped particle entry	[30]
MERS pseudotyped particle entry (HUH7 tox counter screen)	[30]
** Viral Entry **	
Spike-ACE2 protein–protein interaction (alphalisa)	[31]
Spike-ACE2 protein–protein interaction (truhit counter screen)	[31]
Spike-ACE2 protein–protein interaction (qd)	[32]
ACE2 enzymatic activity	[33]
TMPRSS2 enzymatic activity	[34]
3CL enzymatic activity	[35]
** Viral Replication **	
RdRp enzymatic activity	[36] *
HEK 293 cell line toxicity	[37] *
** Human Cell Toxicity **	
Human fibroblast toxicity	[38] *

* = Variant Therapeutic Assay Variant Studies (nih.gov).

## Data Availability

The data in the report were derived from publicly available studies.

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
