# Peer review of "Screening Drugs for Broad-Spectrum, Host-Directed Antiviral Activity: Lessons from the Development of Probenecid for COVID-19"

_viruses, 2023, doi:10.3390/v15112254_

Round 1

Reviewer 1 Report

Comments and Suggestions for Authors

Tripp et al., in the opinion manuscript "Screening Drugs for Broad-Spectrum, Host-Directed Antiviral Activity: Lessons from the Development of Probenecid for COVID-19" have addressed an important aspect of addressing targeting host factors for antiviral approach. 

I have only two comments.

1) Authors should include examples of studies targeting host factors. A table of a paragraph will be great. 

2) Authors should elaborate the advantages of targeting host factors including reduced rates/chances of drug resistance.  

Author Response

Reviewer 1

1) Authors should include examples of studies targeting host factors. A table of a paragraph will be great.

  • We have included Table 1, offering examples of targeting host factors.

2) Authors should elaborate the advantages of targeting host factors including reduced rates/chances of drug resistance.

  • We have noted the major advantage of targeting host factors in the introduction.

Reviewer 2

In this submission, the authors compare direct-acting antiviral (DDA) development with host-directed antivirals (HDA) for SARS-CoV-2. The researchers stressed that HDAs that inhibit virus replication by inhibiting a host gene or pathway are often missed. Fewer HDAs have been tested in preclinical or clinical studies. The manuscript is, in general, well-written and easy to follow. I have no significant concerns regarding the author’s opinion. There is a small question.

How can more clinical data for HDA drugs be obtained to determine whether the drugs are effective against viruses?

  • We thank the reviewer. The reviewer's question is central to this paper, i.e., why DDAs are prevalent over HDAs. We note throughout the paper and emphasize that several barriers exist to discovering HDAs, one being the need for in-depth knowledge needed to understand virus-host interactions and their significance to virus replication. Another is the lack of a suitable model, as HDAs target human genes/factors, and the animal models may not truly represent the host.

The revised manuscript is 2156 words.

Reviewer 2 Report

Comments and Suggestions for Authors

Comments to the Author 

In this submission, the authors compare direct-acting antiviral (DDA) development with host-directed antivirals (HDA) for SARS-CoV-2. The researchers stressed that HDAs that inhibit virus replication by inhibiting a host gene or pathway are often missed. Fewer HDAs have been tested in preclinical or clinical studies. The manuscript is, in general, well-written and easy to follow. I have no significant concerns regarding the author’s opinion. There is a small question.

How can more clinical data for HDA drugs be obtained to determine whether the drugs are effective against viruses? 

Author Response

(The authors gave the same response as above.)
